# Development of Epoxy and Urethane Thermosetting Resin Using *Chlorella* sp. as Curing Agent for Materials with Low Environmental Impact

**DOI:** 10.3390/polym15132968

**Published:** 2023-07-06

**Authors:** Kohei Iritani, Akihito Nakanishi, Rinka Nihei, Shiomi Sugitani, Takashi Yamashita

**Affiliations:** 1Department of Applied Chemistry, School of Engineering, Tokyo University of Technology, 1404-1 Katakuramachi, Hachioji, Tokyo 192-0982, Japan; 2Research Center for Advanced Lignin-Based Materials, Tokyo University of Technology, 1404-1 Katakuramachi, Hachioji, Tokyo 192-0982, Japan; 3School of Bioscience and Biotechnology, Tokyo University of Technology, 1404-1 Katakuramachi, Hachioji, Tokyo 192-0982, Japan; 4Graduate School of Bionics, Computer and Media Sciences, Bionics Program, Tokyo University of Technology, 1404-1 Katakuramachi, Hachioji, Tokyo 192-0982, Japan

**Keywords:** cell-plastics, bioplastics, thermosetting resin, epoxy resin, urethane resin, green alga

## Abstract

In the current system, the disposal of plastic materials causes serious environmental pollution such as the generation of carbon dioxide and destruction of the ecosystem by micro-plastics. To solve this problem, bioplastics, biomass and biodegradable plastics have been developed. As part of our research, we have developed novel bioplastics called “cell-plastics”, in which a unicellular green algal cell serves as a fundamental resource. The production of the cell-plastics would be expected to reduce environmental impact due to the usage of a natural product. Herein, to overcome the mechanical strength of cell-plastics, we used thermosetting epoxy and urethane resins containing *Chlorella* sp. as the green algae. We successfully fabricated thermosetting resins with a *Chlorella* sp. content of approximately 70 wt% or more. IR measurements revealed that the chemical structure of an epoxide or isocyanate monomer mixed with *Chlorella* sp. was modified, which suggests that the resins were hardened by the chemical reaction. In addition, we investigated the effect of thermosetting conditions such as temperature and compression for curing both resins. It was revealed that the Young’s moduli and tensile strengths were controlled by thermosetting temperature and compression, whereas the elongation ratios of the resins were constant at low values regardless of the conditions.

## 1. Introduction

Bioplastics have attracted intense interest in the context of the construction of a sustainable society [1,2,3,4,5]. They are classified as biomass [6,7,8,9] and biodegradable plastics [10,11,12,13,14]. The former are fabricated from biomass resources present in nature through various production processes such as fermentation, saccharification and polymerization. This would be useful for the construction of a carbon cycle system because CO_2_ in the atmosphere is used for the growth of biomass. On the other hand, biodegradable plastics are decomposed into CO_2_ and H_2_O in the environment. Therefore, it is expected that biodegradable plastics have the potential to solve the environmental destruction caused by plastics leaked into the environment. However, there are some problems related to bioplastics: the fabrication of the biomass for plastics requires a lot of energy and money because it is a multi-step production process [6,7], and biodegradable plastics have the disadvantage of poor long-term storage owing to their decomposition in the environment. In addition, not all biomass plastics are necessarily biodegradable, and the raw material for biodegradable plastics is often produced from petroleum; that is, the number of plastics having the properties of biomass and biodegradable plastics is small and there are few examples [15,16,17,18,19,20]. For instance, poly lactic acid is one of the examples of the biomass plastics with biodegradability [17,18,19,20]. It has been produced industrially, and it is becoming more widespread. However, there are some problems such as the low glass transition temperature and the heat resistance of approximately 60 °C [20], and the necessity of temperature conditions at approximately 60 °C for biodegradation [17].

As an alternative method, we have developed a novel bioplastic called cell-plastic, in which a green alga is used as a raw material without any transformation processes [21,22,23,24,25,26,27]. Green alga is distributed in oceans and lakes all over the word with ca. 50 Pg-C/year, although this distribution is uneven [28,29]. Among the green algae, *Chlorella* sp. is used as the biomass resource because it is known that *Chlorella* sp. is decomposed by a naturally occurring virus [30,31,32] called *Chlorella* virus, although it has low biodegradable efficiency. In other words, cell-plastics have the potential to decompose naturally while overcoming the problem of storage stability. In addition, artificial cultivation of *Chlorella* sp. using a photobioreactor has also been successful at 22.8 g/(m^2^·day) [33,34]. In this paper, we call *Chlorella* sp. the cell. For the fabrication of cell-plastics, we have used a polymer as a matrix because the mechanical properties of an aggregation of the cells are quite low to use it in daily life. In the past, the biodegradable polymers such as polybutylene succinate [22] and polyvinyl alcohol [25] were used as the matrix. The mechanical properties of a composite film of the matrix and cell decreased depending on the containing ratio of the cell, indicating that the composite films were destroyed by external loads due to the low interactions between the matrix and the cell. To solve this problem, we planned to fabricate a thermosetting resin using a chemical reaction between the cell and an organic material. In general, because the cell wall of *Chlorella* sp. is composed of carbohydrate, it would be expected that a hydroxyl group is present in the cell wall. Therefore, we hypothesized that a molecule having a functional unit to react with the hydroxyl group could be used as a raw material for a thermosetting cell resin. In this study, we proposed an investigation of the reactivity and thermosetting property of a mixture of a bifunctional monomer and cell. As the functional unit, an epoxide or isocyanate group was used due to the high reactivity with the hydroxyl group for an epoxy [35,36,37] or urethane resin [38,39], respectively. To this end, we chose 1,4-butanediol diglycidyl ether (BDE) or 1,3-bis(isocyanatomethyl)cyclohexane (BIC) as the monomer because these molecules would be relatively easy to make a composite with the cell because of their liquid state at room temperature in the atmosphere (Figure 1a,b). Moreover, in the case of the mixture of BDE and the cell, 1,8-diazabicyclo [5.4.0]undec-7-ene (DBU) was used as a basic catalyst to promote thermosetting (Figure 1c). Figure 1d,e show expected chemical reactions of the epoxide and isocyanate monomer with the hydroxyl group, respectively. We successfully fabricated the thermosetting cell resins with the cell containing between approximately 70 and 90 wt%, while the composites with a small cell content of 50 wt% or less did not harden, indicating that materials with high biomass content could inevitably be constructed. In addition, IR measurements revealed that the chemical structures of the monomers changed before and after the thermosetting process. Surprisingly, for the epoxy/cell resin, it was found that the resins cured without using DBU. To investigate the effect of curing conditions for the fabrication of thermosetting cell resins, the epoxy/cell and urethane/cell resins were molded by controlling the heating temperature and compression, and their mechanical characteristics were evaluated. As a result, it was clarified that the Young’s moduli and tensile strengths of both resins improved depending on increasing the pressure and temperature, and that brittle resins were obtained at 180 °C due to hardening of *Chlorella* itself. These results open up a route to fabricate sustainable and functional cell-plastic materials.

## 2. Materials and Methods

### 2.1. Materials

All commercially available reagents were procured from Tokyo Chemical Industry Co., Ltd. (Tokyo, Japan) and Wako Pure Chemical Industries, Ltd. (Osaka, Japan) and used without further purification. *Chlorella* sp. was obtained from Yasashisa Kyoto-kan and used without any chemical or physical treatment such as additional drying and grinding. The size of the powder particles was from approximately 1 to 100 micro meters [25].

### 2.2. Fabrications of Resins

For both resins, the monomer and Chlorella were mixed twice for 1 min at 2000 rpm using a mixer (ARE-310, Thinky Co., Ltd., Tokyo, Japan). For the fabrication of epoxy–cell resins, BDE and DBU (less than 10 wt%) were added to a powder of *Chlorella* sp. to prepare a mixture of *Chlorella* sp. of 47, 69, 76, 80 and 82 wt% with the monomer. Subsequently the mixture was placed on a polyimide film (Upilex, Ube Industries, Ltd., Tokyo, Japan) to prepare a plate-like structure of the thermosetting resin through heat compression at 120 °C and 20 MPa for 30 min. Urethane–cell resins with *Chlorella* sp. of 50, 83, 87, 89 and 95 wt% were prepared through a similar method to the epoxy–cell resins.

### 2.3. IR Measurements

IR spectra were measured using an instrument (IR Affinity-1S, Shimadzu Co. Ltd., Kyoto, Japan) and an ATR method. The sample was mechanically crushed before measurement. Subsequently, it was placed on a measuring plate and was sandwiched with a fixture to obtain spectra.

### 2.4. Tensile Tests

We used a tensile strength tester (EX-SX, Shimadzu Co. Ltd., Kyoto, Japan) to determine the Young’s modulus, tensile strength and elongation at break of the resins. To fit into the tensile strength tester, the films were typically cut to 10 mm × 150 mm using a cutter. The thickness of the test piece was measured using a film thickness meter before every measurement. The initial distance between the fixtures of the tester was typically set to 10 mm. The crosshead rate was set to 1 mm/min. The Young’s modulus *E* of the film was calculated by using Equation (1):*E* = (*W*/*A*)/(*X*/*L*)(1)
where *W*, *A*, *X* and *L* are the load on the sample for the tensile test, area of cross section, displacement during tensile test and the initial distance between the fixtures, respectively. The values of *W* and *X* were detected through tensile strength tests. The area of cross section *A* was calculated as the product of the width (typically 10 mm) and the thickness measured using a film thickness meter. The length *L* was typically set to 10 mm. In addition, stress and strain are given by *W*/*A* and *X*/*L*, respectively. The stress was plotted as a function of strain for the tensile strength tests. Young’s modulus was derived using the slope of an initial straight line approximated via the least square method. The tensile strength was defined as the maximum stress value. Moreover, the value of elongation at break was determined by *X*/*L* with zero load. The values of the mechanical properties of each film were the averages of at least 4 test pieces.

### 2.5. SEM Observations

Before SEM imaging, each resin was ion-coated by using an ion coater (IB-2, Eiko Co., Tokyo, Japan). SEM images were acquired by using an instrument (JSM-6060LV SEM, Japan Electron Optics Laboratory Co., Ltd., Tokyo, Japan). The accelerating voltage was set to 15 kV during the observations. The SEM images were obtained as secondary electron images (SEI).

### 2.6. TG and DTA Measurements

Differential thermal analysis (DTA) was carried out using an instrument (DTA-60, Shimadzu Co. Ltd., Kyoto, Japan) at a heating rate of 5 °C/min under nitrogen gas (100 mL/min). To suppress the influence of water, preheating was performed from room temperature to 100 °C and the temperature was kept at 100 °C for 10 min. Subsequently, after cooling down to 50 °C or less, the measurement was performed at 600 °C.

## 3. Results and Discussion

### 3.1. Preparations of Thermosetting Cell Resins

To decide the preferable mixing ratio of the cell, mixtures of the cell and BDE with or without DBU as the basic catalyst were prepared by manually mixing at the containing ratio of the cell shown in Table 1. Thermosetting epoxy resins were prepared via heat compression at 150 °C with 20 MPa for 60 min. In the case of 47 and 82 wt% cell content, the mixture formed a brittle plate-like structure with not enough strength to evaluate the mechanical property. For the mixture with cell content between 69 and 80 wt%, relatively hard plate-like materials were obtained. At 80 wt% of cell content, the mixture hardened without DBU. The thermosetting urethane resin was formed through a similar method without the addition of DBU. In Table 2, the mixing ratios of the resources are shown. A mixture of 50 wt% cell content did not form cured resin, whereas at the cell contents between 75 and 89 wt%, cured plate-like structures were fabricated. In the case of the 95 wt% cell mixture, a brittle plate was obtained. It was revealed that approximately 70 wt% cell content would be required for the fabrication of the thermosetting cell resin. We decided to examine the curing conditions with the mixing ratio of 80 wt% of cell content.

In addition, we demonstrated IR measurements for investigation of the chemical structural changes of the monomers. Figure 2 exhibits IR spectra of the thermosetting epoxy– and urethane–cell resin before and after heat compression, respectively. In the case of the epoxy–cell resin, although a signal at 910 cm^−1^ corresponding to the stretching vibration of C-O in the epoxy group was observed in the mixture before heating, it disappeared after resinization (Figure 2a,b). For the urethane–cell resin, in general, signals of a urethane bond, which appears from the reaction of isocyanate with the hydroxyl group, are detected at approximately 1640 cm^−1^ for the stretching vibration of C=O and 1510 cm^−1^ for the deformation vibration of C-N. Unfortunately, it was difficult to evaluate the chemical reaction of the isocyanate monomer because the IR spectrum of cell also shows signals at similar positions to those of the urethane bond (Figure 2c,d). Note that the transmittance ratio of the stretching vibration of C=N (ca. 2256 cm^−1^) in isocyanate with respect to that of aliphatic C-H (ca. 2918 cm^−1^) was quite low compared to the ratio of the monomer, which indicated that the chemical structure of the monomer in the urethane resin had changed. Based on IR measurements, it was considered that the heating time until the reaction completed was sufficient for 1 h in both resins.

### 3.2. Evaluations of Mechanical Properties of Thermosetting Cell Resins

To evaluate the influence of compression and temperature on the curing of thermosetting epoxy– and urethane–cell resins, tensile tests were conducted using heat-compressed samples at 120, 150 or 180 °C and 5, 10 or 20 Mpa, respectively. Table 3 shows the Young’s moduli, tensile strengths and elongations at break of the materials. Note that data for the resins formed with low compression (5 Mpa) for the epoxy/cell resin and with high temperature (180 °C) for both resins were not obtained because it was difficult to perform their tensile tests because test pieces were not prepared due to toothier brittleness. In addition, Figure 3 shows SEM images of the cross sections and surfaces of both resins. In the case of the epoxy/cell resins formed at 120 °C, the spherical structure corresponding to the particle of *Chlorella* was observed in the resin with 5 Mpa compression. On the other hand, for the cured resins formed with higher compression, larger structures compared to *Chlorella* were observed. In particular, the particle structure was almost imperceptible at 20 Mpa compression, which indicated coalescence of the particles, likely due to chemical reactions between *Chlorella* and the monomer. Moreover, it was revealed that the Young’s modulus and tensile strength of the resin formed with 20 Mpa compression were greater than that with 10 Mpa compression. These results indicated that the structure of *Chlorella* is destroyed by heat and compression during the formation of the resin. At 150 °C, a similar tendency in appearance and in mechanical properties to the resin formed at 120 °C was observed, although the structure of resin formed with 20 Mpa compression exhibited lower uniformity than that at 120 °C with 20 Mpa. Note that the mechanical properties of 150 °C were greater than those of 120 °C, which indicated that the reaction was accelerated by high temperature. The elongations at the break were detected to be 3.0% or less in all resins. For the resins formed at 180 °C, although partial coalescence of the particles was observed, the spherical *Chlorella* was often observed, even in the resin with 20 Mpa compression. It was hypothesized that at 180 °C, the proteins contained in *Chlorella* denatured before its structural modification and chemical reaction with the monomer.

For the urethane/cell resins formed at 120 °C, partial coalescence of the particle of *Chlorella* was observed in all compression values based on SEM observations. Their Young’s moduli and tensile strengths slightly increased depending on the compression with comparison to those of the epoxy/cell resins formed under the same conditions. On the other hand, at 150 °C, the mechanical properties were significantly increased with increasing the compression values. Based on SEM observations, it was revealed that the larger structures due to the coalescence of *Chlorella* were formed in the resin with 20 Mpa compression, similar to the epoxy/cell resin formed at 120 °C with 20 Mpa. It was found that the deformation of *Chlorella* due to curing caused a significant increase in mechanical properties. In addition, at 180 °C, the urethane/cell resin was cured with 5 Mpa compression and its internal structure exhibited partial coalescence of *Chlorella*, whereas at 10 and 20 Mpa, the resins did not cure sufficiently and the spherical *Chlorella* often remained, although partial coalescent structures were observed. As for elongation at break, the average values of the urethane/cell resins were 4.0% or less, similar to those of the epoxy/cell resins.

We successfully fabricated the thermosetting resins using *Chlorella* as the curing agent. Note that the mechanical properties of general epoxy resins and polyurethane materials are as follows [40]: Young’s moduli, tensile strengths and elongations at break are 2400 Mpa, 27–89 Mpa and 3.0–6.0%, respectively, for the epoxy resin, and 70–690 Mpa, 1–69 Mpa and 100–10,000%, respectively, for polyurethane. In general, the epoxy resins show high mechanical strength, although they have low elongation because they have cross-linked structures with high density [41,42,43]. With regard to the epoxy/cell resins, it was considered that the resins using *Chlorella* as the curing agent exhibited similar mechanical behavior to the general resins because cross-linked structures were formed due to the chemical reaction between BDE and *Chlorella*. In addition, typical stress–strain curves of the epoxy/cell resins under each curing condition are shown in Figure 4a. It was found that the resins were broken In the region of elastic deformation, showing similar mechanical behavior to the general resins. On the other hand, it is known that the general polyurethane material has high elongation, and its elongation is controlled depending on the cross-linking density due to the chemical structure of polyol as the curing agent [44,45,46]. For the urethane/cell resins, because *Chlorella*, which would be thought to contain more hydroxyl group, was used instead of the polyol, it is considered that their cross-linking density increased and the elongation ratio decreased significantly. Moreover, from typical stress–strain curves of the urethane/cell resins as shown in Figure 4b, it was revealed that the resin breakage occurred without inelastic change, unlike the general polyurethane materials.

In addition, we conducted thermogravimetric (TG) and differential thermal analyses (DTA) of the resins. Their TG-DTA curves are shown in Figure 5. It was revealed that the mass losses and DTA curves of the resins were almost comparable to that of cell in both cases, although ash content at 600 °C increased more than that of cell. In addition, in the case of the thermosetting epoxy–cell resin, the 10 wt% mass loss temperature was evaluated to be 264 °C, whereas that of cell was 248 °C. The former slightly increased more than the latter, suggesting that heat resistance was slightly improved by heat curing.

## 4. Conclusions

For the development of low-environmental impact materials for use in sustainable society, we fabricated thermosetting epoxy and urethane resins using *Chlorella* sp. as the hardened material in anticipation of hardening through the chemical reaction with a hydroxyl group in the cell wall. As a result of examining the curing conditions, it was clarified that these resins cured only at a cell content of between approximately 70 and 90 wt%. In addition, IR measurements revealed that the chemical structures of the epoxide or isocyanate monomer changed with hardening of the resin. Moreover, we successfully controlled the mechanical properties of the epoxy/cell and urethane/cell resins using temperature and compression in the thermosetting curing process. These resins had increased mechanical strength with increasing temperature and compression, although the curable resin was not molded at 120 °C and 5 MPa for the epoxy/cell resin or at 180 °C for both resins. In addition, SEM observations revealed that the internal structures of the hardened resins exhibited larger massive structures compared to the *Chlorella* particle, likely due to the coalescence of *Chlorella*. Based on the findings obtained from this research, we aimed to fabricate thermosetting cell resins using a polymer matrix containing an epoxy and isocyanate group to create thermosetting cell resins with higher mechanical strength and multifunctionality. These results provide insight into how the chemical structures of matrices and the mixing ratio of components affect the production of thermosetting cell materials and bring us closer to environmentally friendly plastics to support a sustainable society in the near future.

## Figures and Tables

**Figure 1 polymers-15-02968-f001:**
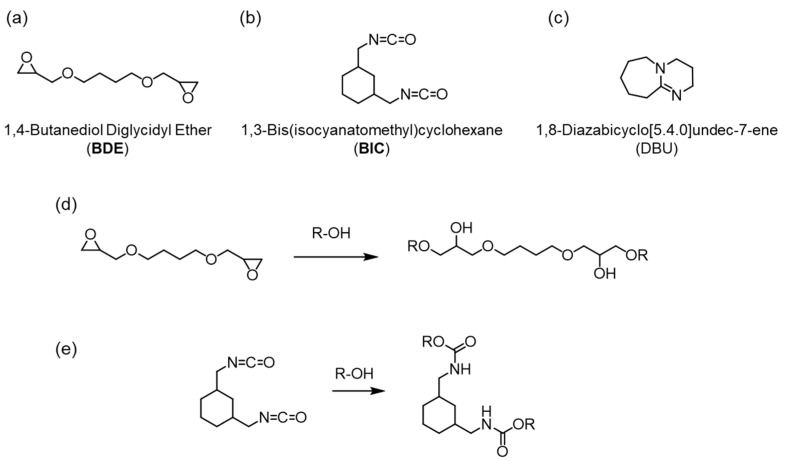
Chemical structures of BDE (**a**), BIC (**b**) and DBU (**c**), and chemical reactions of BDE (**d**) and BIC (**e**) with the hydroxyl group.

**Figure 2 polymers-15-02968-f002:**
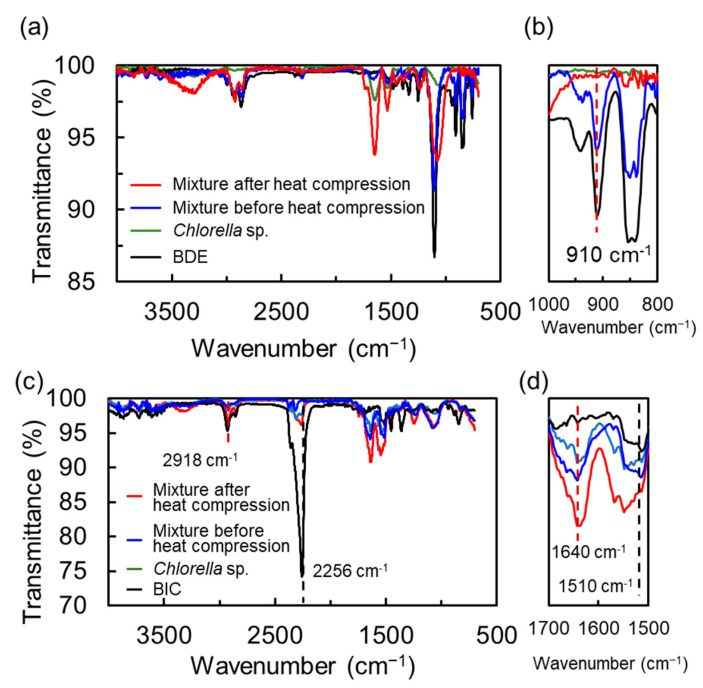
IR spectra of the thermosetting epoxy–cell (**a**,**b**) and urethane–cell (**c**,**d**) resins. Spectra enlarged for each wavelength region are shown in (**b**,**d**). The spectra of the mixture of monomer and *Chlorella* sp. After and before heat compression, *Chlorella* sp. and monomer are shown by red, blue, green and black lines.

**Figure 3 polymers-15-02968-f003:**
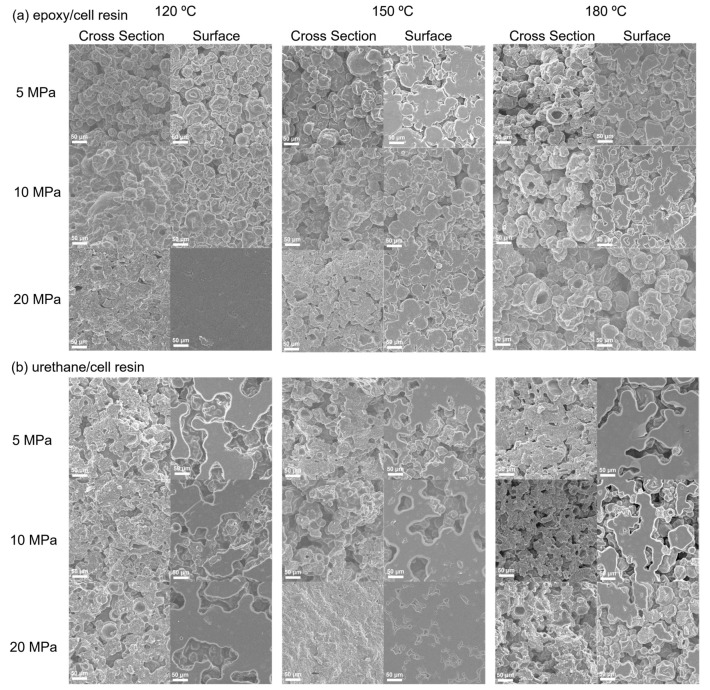
SEM images of the surface and cross section of the thermosetting epoxy– (**a**) and urethane–cell (**b**) resins.

**Figure 4 polymers-15-02968-f004:**
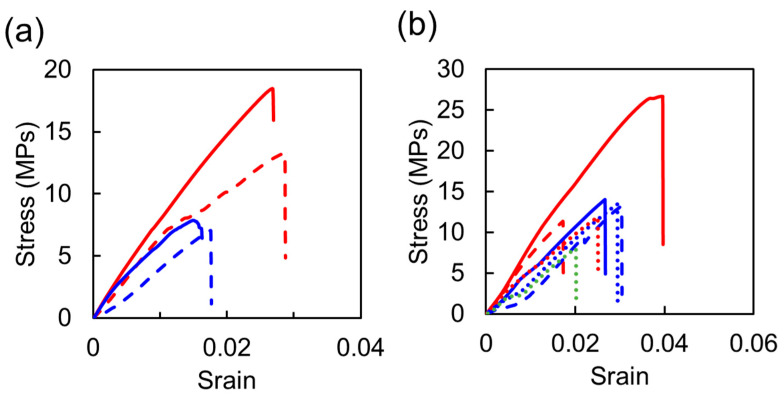
Typical stress–strain curves of the epoxy/cell (**a**) and urethane/cell (**b**) resins. The blue, red and green lines show the data of the resins formed at 120, 150 and 180 °C, respectively. The solid, dashed and dotted lines show the data of the resins formed at 20, 10 and 5 MPa, respectively.

**Figure 5 polymers-15-02968-f005:**
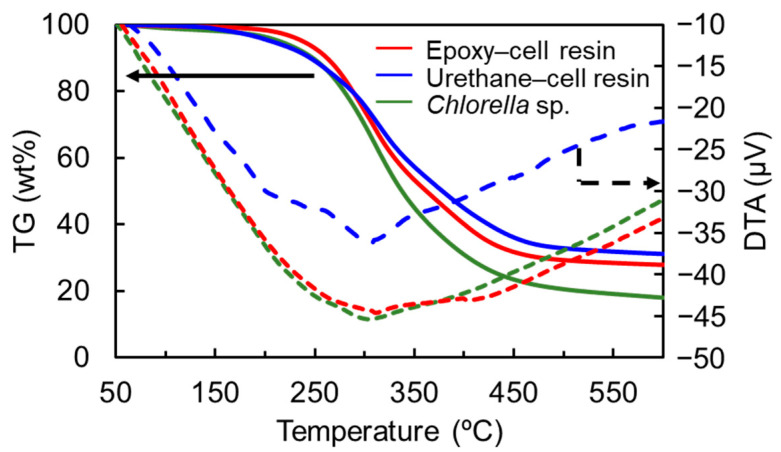
TG (solid lines) and DTA (dashed lines) curves of the thermosetting epoxy– (red) and urethane–cell resins (blue) and *Chlorella* (green). Solid and dashed arrows indicate TG and DTA data, respectively.

**Table 1 polymers-15-02968-t001:** Preparations of the thermosetting epoxy–cell resins.

Weight of Content (g)			Containing Ratio (wt%)			Curability
BDE	DBU	*Chlorella* sp.	BDE	DBU	*Chlorella* sp.	
3.02	0.35	3.04	47	6	47	Brittle
1.02	0.33	3.03	23	8	69	Hardened
0.61	0.34	3.06	15	9	76	Hardened
0.43	0.33	3.00	11	9	80	Hardened
2.03	-	8.03	20	-	80	Hardened
0.34	0.34	3.04	9	9	82	Brittle

**Table 2 polymers-15-02968-t002:** Preparations of the thermosetting urethane–cell resins.

Weight of Content (g)		Containing Ratio (wt%)		Curability
BIC	*Chlorella* sp.	BIC	*Chlorella* sp.	
1.61	1.61	50	50	No Curability
2.03	8.03	20	80	Hardened
0.51	2.51	17	83	Hardened
1.53	10.7	13	87	Hardened
1.32	10.6	11	89	Hardened
0.15	3.00	5	95	Brittle

**Table 3 polymers-15-02968-t003:** Young’s modulus, tensile strength and elongation at break of the thermosetting epoxy– and urethane–cell resins.

Resin	Temperature (°C)	Pressure (MPa)	Young’s Modulus (MPa)	Tensile Strength (MPa)	Elongation at Break (%)
Epoxy	120	5	- *^a^*	- *^a^*	- *^a^*
		10	390 ± 40	6.3 ± 0.8	2.1 ± 0.3
		20	610 ± 90	8.9 ± 1.4	2.5 ± 0.9
	150	5	- *^a^*	- *^a^*	- *^a^*
		10	620 ± 70	11 ± 3	2.6 ± 0.7
		20	710 ± 180	16 ± 4	3.0 ± 0.4
	180	5	- *^a^*	- *^a^*	- *^a^*
		10	- *^a^*	- *^a^*	- *^a^*
		20	- *^a^*	- *^a^*	- *^a^*
Urethane	120	5	600 ± 55	11 ± 2	2.5 ± 0.4
		10	680 ± 44	13 ± 2	2.5 ± 0.6
		20	760 ± 80	16 ± 2	2.9 ± 0.4
	150	5	480 ± 80	10 ± 3	2.4 ± 0.2
		10	610 ± 120	8.4 ± 2.9	1.6 ± 0.5
		20	800 ± 80	18 ± 6	4.0 ± 1.0
	180	5	510 ± 60	9.6 ± 3.2	2.3 ± 0.6
		10	- *^a^*	- *^a^*	- *^a^*
		20	- *^a^*	- *^a^*	- *^a^*

*^a^* Unmeasurable data because test pieces were not prepared due to the brittleness for the tensile test.

## Data Availability

Not applicable.

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
