# Peer review of "Development of Epoxy and Urethane Thermosetting Resin Using Chlorella sp. as Curing Agent for Materials with Low Environmental Impact"

_polymers, 2023, doi:10.3390/polym15132968_

Round 1

Reviewer 1 Report

This article is very interesting, but there are some notes from me as follows:

  1. For young modulus data, please also include the experimental design. It is also necessary to compare commercial epoxy and urethane as controls.
  2. Is the application of the adhesive product you produce for exterior or interior use?
  3. It is necessary to add applications for using this product, for example, for plywood or other composite boards, to obtain data related to its bonding properties.
  4. To fabricate epoxy-cell resins, BDE and DBU (less than10 wt%) were added 91 to a powder of Chlorella sp. to prepare a mixture of Chlorella sp. of 47, 69, 76, 80, and 82 92 wt% with the monomer. What is the basis for the determination?
  5. Urethane-cell resins with Chlorella sp. of 50, 83, 87, 89, and 95 wt%. What is the basis for the determination?
  6. Prepare a plate-like structure of the thermosetting resin by heat 94 compression at 150 °C with 20 MPa for 60 min. What is the basis for the determination?
  7. Based on the title, you developed this product to reduce environmental impact. How significant is the contribution, and how do you ensure the resulting effect can reduce negative environmental impacts?
  8. In the introductory section, it is necessary to include data on the amount and distribution of chlorella; this is to ensure the availability of raw materials when it is developed into a commercial product

Reviewer 2 Report

 In this manuscript, the authors prepared novel type of thermosetting resins using Chlorella sp. powder as one of components. The concept and results are interesting, but the manuscript needs some improvement before publication. I think that the manuscript can be accepted for publication after minor revision.

I suggest a few comments as follows:

1.    Title: I suggest the replace of a few words in the title for readers' better understanding: from “Epoxy or Urethane Thermosetting Resin” to “Epoxy and Urethane Thermosetting Resins”.

2.    Materials and Methods: Information for sample preparation is very important to keep the reproducibility of the experiments. I recommend the authors to add more information in the Materials and Methods section. For example, 1) the authors used powder form of Chlorella sp., but how did they prepare the powder, i.e. what is the drying method, did they grind it or not, and what is the size of the powder particles? 2) how did the authors mix the monomer with Chlorella powder? This information is very important to discuss the dispersity and cavity of the dried cell in the materials.

3.    Line 105, Page 3: The authors describe the explanation of the Equation (1), and defined W as the “sample mass”. Is it true? Why is the sample mass needed to calculate Young’s modulus? It may be a typing mistake.

4.    Lines 164-166, Page 5: The authors described “the values of young’s modulus and tensile strength were substantially constant”. However, I think that the strength of the epoxy-cell resin containing 69% cell is much lower than other two epoxy resins. The reason should be described. Relating to this mechanical property, the authors are recommended to compare SEM images of the epoxy-cell resins with different cell contents.

Reviewer 3 Report

Some methods should be further explained in the case of IR spectrum. What were the conditions of the equipment during the measurements?

The tensile test could also be better explained. How was the dispersion of the results obtained from them? It seems to me that three replicates are too few for this type of test.

Also, SEM measurement should be better explained. What were the conditions used during the micrographics acquisitions?

In the methods section, the range of temperature studied in the DTA should also be indicated.

Only two sentences are used to explain the results of figure 3. This is really few, it’s like this figure and results have no importance, so…why did the authors carry out this test?

In line 169-171 the author give reasons why I think the manuscript does not have scientific soundness enough.

I see a lot of doubts when I read the manuscript as lot of expressions such as "approximately, randomly, seem to be…" there are few affirmative statements.

The scientific quality of the explanations given in the discussion is low. No citations or references are used in order to make stronger the discussion part of the manuscript. So, this section lacks scientific soundness. The authors must improve this aspect if they want the manuscript to be considered to be published in Polymers journal.

English can be improved but is not a critical aspect

Round 2

Reviewer 3 Report

The manuscript has gained sufficient scientific soundness with the suggestion addressed.